# A combined approach for single-cell mRNA and intracellular protein expression analysis

Johan Reimegård[1,2,3,7], Marcel Tarbier[4,7], Marcus Danielsson [3,5,7], Jens Schuster [5], Sathishkumar Baskaran[5], Styliani Panagiotou[5], Niklas Dahl [5], Marc R. Friedländer[4] & Caroline J. Gallant [5,6 ✉]

Combined measurements of mRNA and protein expression in single cells enable in-depth analysis of cellular states. We present SPARC, an approach that combines single-cell RNA-sequencing with proximity extension essays to simultaneously measure global mRNA and 89 intracellular proteins in individual cells. We show that mRNA expression fails to accurately reflect protein abundance at the time of measurement, although the direction of changes is in agreement during neuronal differentiation. Moreover, protein levels of transcription factors better predict their downstream effects than do their corresponding transcripts. Finally, we highlight that protein expression variation is overall lower than mRNA variation, but relative protein variation does not reflect the mRNA level. Our results demonstrate that mRNA and protein measurements in single cells provide different and complementary information regarding cell states. SPARC presents a state-of-the-art co-profiling method that overcomes current limitations in throughput and protein localization, including removing the need for cell fixation.

[1] National Bioinformatics Infrastructure Sweden, Uppsala University, Uppsala, Sweden. [2] Department of Cell and Molecular Biology, Uppsala University, Uppsala, Sweden. [3] Science for Life Laboratory, Uppsala University, Uppsala, Sweden. [4] Science for Life Laboratory, Department of Molecular Biosciences, The Wenner-Gren Institute, Stockholm University, Stockholm, Sweden. [5] Department of Immunology, Genetics and Pathology, Uppsala University, Uppsala, Sweden. [6] Present address: 10x Genomics, Stockholm, Sweden. [7] These authors contributed equally: Johan Reimegård, Marcel Tarbier, Marcus Danielsson. ✉email: caroline.gallant@10xgenomics.com

Advances in single cell analysis are impacting the scale and resolution at which we investigate biological systems. The majority of these advances are focused on the application of single cell RNA sequencing (scRNAseq). However, transcriptomes may not fully reflect cell states. scRNAseq provides comprehensive snapshots of gene expression, but gene transcription is stochastic and characterized by transcriptional bursts of varying rates and sizes[1], and half-lives of mRNA molecules vary significantly between genes[2]. Sampling from low number of molecules, variable efficiency to convert mRNA to cDNA and PCR amplification bias during sequencing library preparation also contribute to noisy expression data[3].

In contrast to mRNA, proteins are more stable, and typically present in orders of magnitude higher amounts within the cell[2], reducing chance fluctuations of their levels. Moreover, proteins have more direct roles in maintaining cellular functions compared to transcripts. Therefore, we argue that combined mRNA and protein single cell measurement approaches are necessary to better understand cellular states and to decipher regulatory circuits and pathways.

A number of approaches are emerging to simultaneously measure both mRNA and protein at single cell resolution. These include several methods for targeted mRNA and protein detection[4–7]. In contrast, measuring global mRNA enables entirely new studies, such as the impact of regulatory proteins on targets transcriptome-wide. A number of methods including CITE-seq allow for the measurement of global mRNA and a large number of proteins, but are limited to surface proteins[8,9]. One published method has profiled global mRNA and intracellular proteins, but this study was limited to six proteins and required invasive cell fixation[10]. We describe an approach, herein called Single-Cell Protein And RNA Co-profiling (SPARC), which enables measurement of global mRNA and high multiplex, targeted intracellular proteins in single cells (Fig. 1a). mRNA levels were recorded using a modified Smart-seq2 protocol[11] enabling sensitive expression measurements of the full-length transcripts.

The use of Smart-seq2 replaces a targeted, multiplexed qPCR approach described previously[7]. Proteins were measured using multiplex, homogeneous protein extension assays (PEA)[7,12], an affinity-based protein detection method that allows scalable protein detection in fresh cell or tissue lysates without a need for prior fixation.

PEA is a member of a class of proximity-based assays that require two binding events in order to generate a DNA reporter molecule. In turn, the reporter can be detected and quantified using various DNA detection technologies, including quantitative PCR and sequencing. Here, proteins are detected using pairs of antibodies conjugated with oligonucleotides whose free 3′ ends are pairwise complementary. When cognate antibody pairs bind their target protein, the attached oligonucleotides are brought in proximity and can be extended by polymerization to create amplifiable DNA reporter molecules. A multiplex readout is achieved by decoding extension-generated DNA reporters by real-time PCR using primer pairs specific for cognate pairs of antibody conjugates. The requirement for pairwise protein detection ensures high assay quality, similar to i.e., sandwich immunoassays, while allowing for simultaneous measurement of many proteins in each reaction. Moreover, PEA assays do not require fixation of the cell, capture of target proteins on solid supports or wash steps following the addition of the oligonucleotide-conjugated antibody probes to the cell lysate.

With SPARC, we present a powerful approach to quantitatively measure mRNA and protein in single cells and show how protein measurements greatly aid the analysis of gene expression variation, cell states, and cellular regulatory mechanisms. We applied the method to investigate to what extent the amounts of a transcript are predictive of the levels of the corresponding protein in cells at steady-state or undergoing a state-transition. Specifically, we measure mRNA and protein in human embryonic stem cells (hESCs) unperturbed or at fixed time points after induction of directed neuronal differentiation (Fig. 1b). We also investigate the effects of biological inferences, including transcription factor

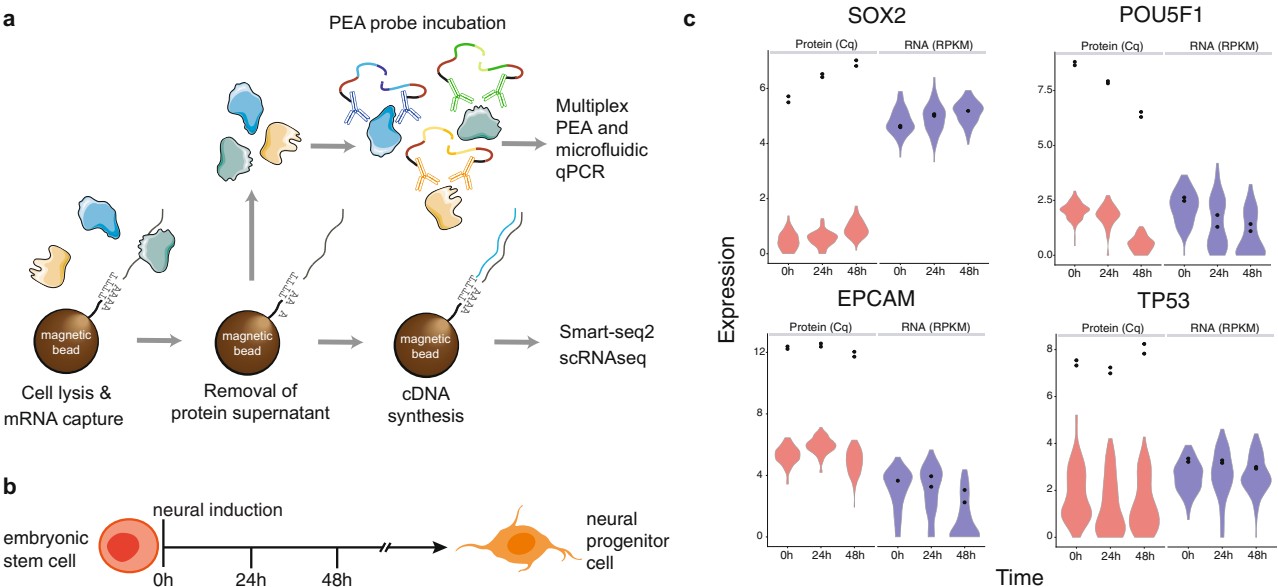

**Fig. 1 SPARC profiles RNA and intra-cellular protein from the same single cells. a** The SPARC procedure. Single cells are isolated and lysed in the presence of oligo-dT conjugated magnetic beads. Following oligo-dT mRNA hybridization, the protein-containing supernatant is removed for subsequent multiplex proximity extension analysis (PEA) and the mRNA is processed using a modified Smart-seq2 approach. (Protein clip art from Servier Medical Art is used under CC BY 3.0 France.) **b** Overview of the cellular model analyzed using SPARC. Human embryonic stem cells were analyzed in culture (0 h) and following directed neural induction (24 h and 48 h). **c** Example mRNA (blue) and protein (red) expression in single cells (violin plots) or replicate 100 cell population control (black dots) measured at 0, 24, and 48 h post neural induction.

regulation, by analyzing scRNAseq data alone or integrated with the targeted protein expression data, and measure the agreement between mRNA and protein expression variation.

## Results

**Single cell mRNA expression data.** The Smart-seq2 scRNAseq method adapted for the SPARC protocol includes a number of steps that differ from the published protocol[11]. The key differences include additional detergents in the cell lysis buffer in order to ensure access to nuclear proteins, the use of oligo-dT conjugated T1 Dynabeads in order to immobilize the poly(A) mRNA fraction, and exclusion of the 72 °C heat step before the reverse transcription reaction to avoid denaturing cellular proteins (Methods). We compared the expression data prepared with the reference Smart-seq2 method (0 h, $n = 67$ cells) and SPARC (0 h, $n = 85$ cells). The cells were sorted and processed for analysis in parallel and the sequencing was done in the same lane of the sequencing flow cell. Only SPARC data was collected for cells during early differentiation (24 h, $n = 76$ cells and 48 h, $n = 86$ cells).

Overall, the scRNA-seq data using SPARC was of high quality and highly comparable to Smart-seq2 (Pearson correlation coefficient rho = 0.90, $N = 51157$) (Supplementary Fig. 1). Some minor differences were observed for numbers of profiled genes ($n = 3308$ for SPARC vs. 3746 for Smart-seq2) or pseudogenes ($n = 113$ vs. 62), the fraction of reads in introns ($n = 0.43$ vs. 0.13) and average length of detected genes (Supplementary Fig. 1). We attribute the greater intron capture with SPARC to lysis conditions providing greater access to nuclear content and the omission of a heating step before the oligo-dT primed reverse transcription. Specifically, we may preferentially access and capture unspliced, nascent mRNA before they mature and are fully coated with RNA binding proteins[13]. Nonetheless, quality control analysis showed that the SPARC mRNA data was of high quality and highly reproducible, allowing us to proceed with the combined mRNA and protein analysis.

**Single cell protein expression data.** We developed an exploratory multiplex PEA panel for single cell analysis in collaboration with Olink Proteomics, involving 96 proteins and focused on intracellular proteins of interest for our investigation. The panel includes proteins across different functional groups related to, for example, pluripotency, neurogenesis, cell cycle phase, and metabolic functions (Supplementary Table 1). The PEA protein panel was developed for application across different cellular models, and therefore not all proteins were expected to be detectable at the single cell level in hESCs. Of the 96 proteins in the panel, 87 proteins had detectable levels in the 100 cell control, and 89 proteins in single cells (Fig. 1c, Supplementary Table 1, Supplementary Fig. 2).

We proceeded to investigate the major sources of variation using PCA for the mRNA and protein expression data sets. Using RNA expression data, the three factors most strongly contributing to variation between the sampled cells were the numbers of detected genes, developmental state, and cell cycle phase. We hypothesize that these significant factors reflect the relationship between the amount of RNA or protein and cell size, with larger cells containing more absolute numbers of molecules[14] (Supplementary Fig. 3a, b). Indeed, we observed higher protein levels as measured by protein sum in G2 versus S or G1 cells for cells FACS sorted by cell cycle phase (Supplementary Fig. 3c). In order to minimize the effects of cell cycle phase, cell size, and mRNA capture efficiency on the mRNA expression variation, the data was normalized for variation related to the cell cycle and the

number of genes detected. Protein data was only normalized per plate and according to background measures (Methods).

To reduce the dimensionality of the data and align the cells along a trajectory based on similarities in their expression patterns, we next performed tSNE and pseudotime analysis. We applied SCORPIUS[15], a single trajectory inference method, to order the dynamic cells along a progression from undifferentiated to a more differentiated state. The analysis was performed separately on normalized mRNA and protein expression data. The cells were largely ordered and grouped according to the sampled time points (0, 24, and 48 h) (Fig. 2a, b) and the cell order was very similar whether the trajectory was determined based on mRNA or protein expression data (Spearman's rho = 0.82, $N = 242$) (Fig. 2c). The results highlight that both the mRNA and protein data generated with SPARC recapitulate the expected dynamic changes of the model cell system.

**Relation of RNA and protein data in single cells in a steady-state condition.** We next investigated the co-expression of mRNA and protein in cells measured at the 0 h time point, purportedly at steady-state. We define cells as being in steady-state when the mean protein or mRNA levels remain relatively constant over several hours[16]. For investigation of correlation, we focused on within gene correlations i.e., the variation of a gene's mRNA and protein concentrations across single cells[17]. We focused on genes where we detected the protein at a level of >3 Cq over background in the 100 cell population control in at least one time point. We found that level of mRNA expression is generally a poor predictor of protein expression in single cells (Pearson correlation coefficient between −0.12 and 0.40, $N = 83$), and that the scaling and the extent of the relationship between mRNA and protein is gene-specific[4] (Fig. 2e and Supplementary Fig. 4).

For genes where we detect both the mRNA and protein in the majority of the cells, the molecules exist in very different dynamic expression ranges—with the mRNA consistently spanning a much broader range of expression levels compared to protein (e.g., EPCAM, CASP3) (Supplementary Fig. 4). We also observed that protein level measurements provide a more stable representation of a gene's expression state with protein expression detected in the majority of cells measured, whereas mRNA expression detection was more variable, with a fraction of cells showing no detectable expression. The latter may reflect temporal variations in RNA abundance or low capture efficiency, a common technical problem in scRNAseq protocols (Supplementary Fig. 4).

We observed a number of genes where mRNA and protein expression are discordant. Specifically, we observed a very low fraction of cells expressing mRNA but a high fraction of cells expressing the cognate protein. A parsimonious explanation for these observations is that we failed to reliably capture, amplify, and sequence the specific mRNAs, an outcome that is common in scRNAseq experiments. To explore this possibility, we isolated total RNA from the same hESC cell line used in the study and performed bulk cell gene expression analysis using TaqMan Gene Expression assays. We targeted three genes showing discordant gene expression (IKBKG, HMOX1, METAP1D) and included EIF4B as a positive control. We clearly detect RNA expression for all analyzed genes, suggesting that mRNA for IKBKG, HMOX1, METAP1D is expressed but was not reliably detected in the single cell experiments.

However, we cannot rule out that the observed discordant mRNA and protein co-expression at the single cell level is due to very short-half lives of the mRNAs in question and therefore absence in many cells, whereas the cognate proteins are relatively stable. Alternatively, but less likely, the protein expression signal

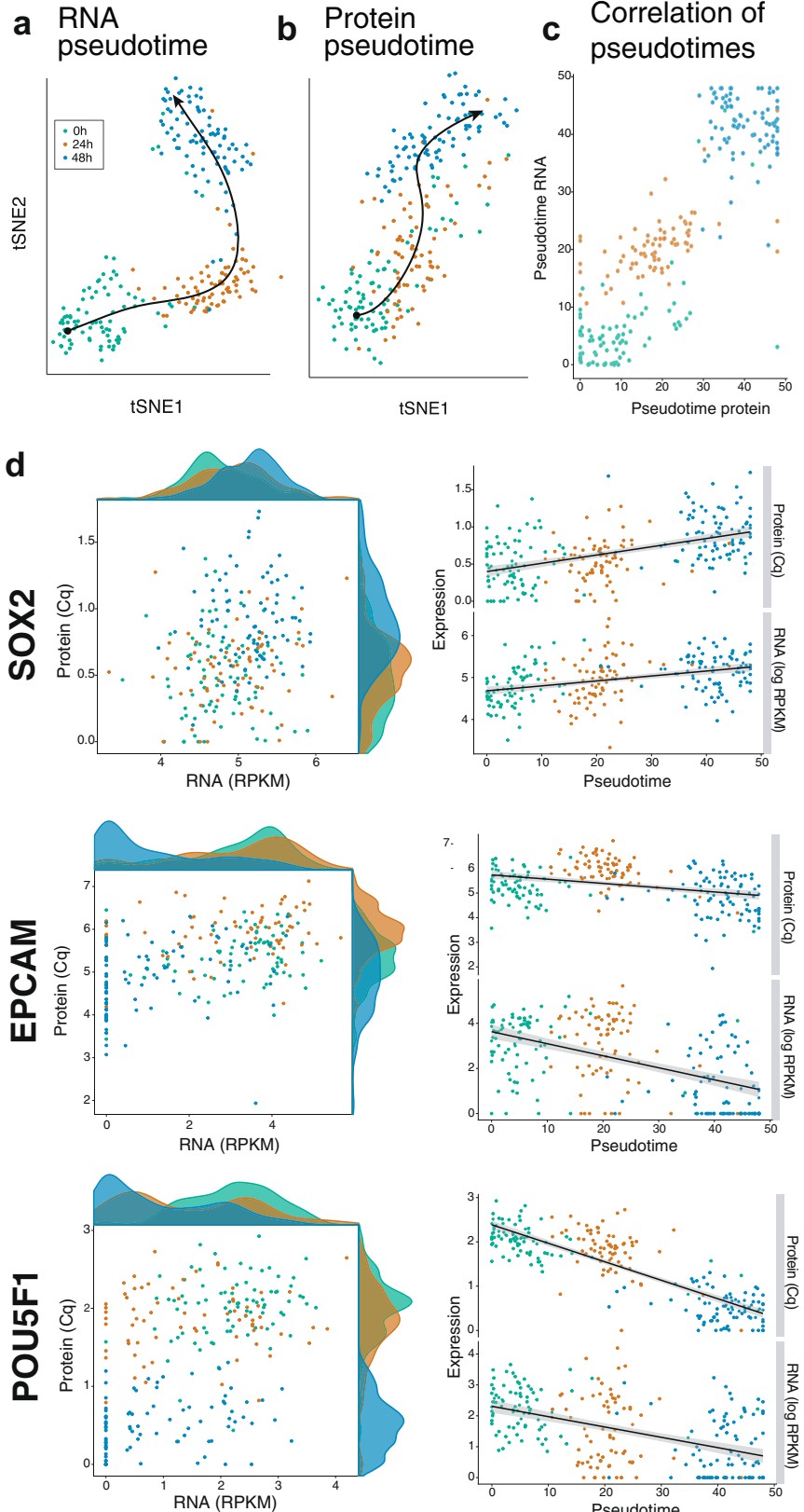

**Fig. 2 mRNA abundance only partially reflects and protein abundance in single cells.** tSNE plot and pseudotime analysis of the **a** normalized mRNA and **b** protein expression data. **c** Comparison of the cell pseudotime-ordering if calculated based on either mRNA or protein expression data. **d** Example mRNA and protein expression data plotted for SOX2, EPCAM, and POU5F1. The data is shown either as RNA expression vs. protein expression scatter plot with respective density plots included on each axis, or as cells ordered along RNA pseudotime with either RNA and protein expression plotted on the y-axis. Both the RNA and protein data are plotted as log2. Color corresponds to time point in all plots 0 h (green), 24 h (orange), and 48 h (blue).

originates from antibody cross-reactivity to homologous gene products. PEA probes employed in this study were confirmed to detect the expected target, and no signal was obtained when tested against a large pool of recombinant proteins.

For some genes measured, the PEA assay was not sensitive enough to detect the protein despite high RNA expression (e.g., PARP1), or the level of detection of the PEA probe was at the limit of detection and therefore qualitative (Table S1). Examples of the latter include the detection of the G2 phase cell cycle markers CCNA2 and AURKA where we only detect expression in predicted G2 cells, the cell cycle phase where they are expected to show peak expression (Supplementary Fig. 2).

**Relation of RNA and protein levels in single cells during dynamic change**. Next, we investigated the agreement of mRNA and protein expression during early time-points of directed neuronal differentiation (Fig. 1b). As with the cells at steady-state, mRNA expression is generally a poor predictor of protein expression. Despite the poor relationship, we were interested to see whether the directional changes of mRNA and protein abundances are in agreement in cells ordered along pseudotime as defined by the mRNA expression data (pseudotime$_{RNA}$) (Fig. 2, Supplementary Fig. 5). To test whether the expression levels for both mRNA and protein level show the same directional changes, we applied a linear model for expression over time for both the RNA log-transformed RPKM values and for protein Cq values. The resulting linear model had a positive slope with a low but significant coefficient of determination ($R^2$) of 0.20 (p-value = $5,1 \times 10^{-4}$, $N = 64$) (Supplementary Fig. 6). The Pearson correlation coefficient was 0.47. These findings indicate that there is a general agreement between gene expression changes at the RNA and the protein level when dynamic processes, such as development, are observed over extended periods of time.

In our data, we observed clear examples of temporal delay of gene expression at the level of mRNA and protein expression. This is well demonstrated by POU5F1, a transcription factor that is rapidly turned off in hESCs upon directed neuroectodermal differentiation[18]. When we order the single cells along pseudotime$_{RNA}$ and plot POU5F1 gene expression, we observed that both POU5F1 mRNA and protein show concordant downward trends in expression (Fig. 2e). However, we note that the protein, but not the mRNA, is detectable in many cells assigned to late pseudotime$_{RNA}$ (corresponding primarily to the cells measured at 48 h). We attribute the time-dependent increase in mRNA–protein expression variability as an effect of gene downregulation and the differential stabilities of mRNA versus protein[19,20]. Long protein half-lives will enable proteins to be present in a cell long after repression of gene transcription.

Overall, the results highlight that while mRNA expression abundances are not predictive of protein abundances at the time of measurement, the differences can be reconciled when we resolve mRNA and protein expression over a (pseudo)temporal scale, and therefore take into account the temporal elements of gene regulation, including the lag times between transcription and translation, and the different half-lives of mRNA and protein molecules. Accordingly, these results also suggest that mRNA and protein measurements in single cells are not redundant but provide different information regarding the cell state at the time of measurement.

**Protein vs. mRNA expression levels correlate better with their trans-regulatory targets**. To date, a major limitation of regulatory network inference analysis is the requirement for very high numbers of replicate observations[21]. Single cell analysis experiments produce hundreds or thousands of independent

measurements and provide an opportunity to use single cell data to power the analysis of causal gene regulatory network analysis. Despite the increase in power, the use of scRNAseq remains a challenge, in part due to the stochastic nature of RNA transcription and the technical limitations of the sequencing protocols.

We were interested in determining whether integrating protein-level expression data with mRNA expression would help decipher gene regulatory networks. We expected an added value of protein measurements due to the observation that single cell mRNA and protein sets are concordant but not redundant. Furthermore, the circumstance that protein expression measurements show lower biological and technical cell-to-cell variation than mRNA measurements, and the expectation that protein expression levels of gene regulatory effectors such as transcription factors should be a closer representation of their functional activity at time of measurement.

To specifically address the question of how protein measurement can complement transcriptomics in analyses of gene regulatory networks, we investigated whether transcription factors can predict expression of their target genes, when measured at the RNA or protein level. We analyzed four important and reliably detected TFs (NOTCH1, POU5F1, SOX2, and TP53) and their predicted targets from three databases, corresponding to 10 sets of TF-target relations, and ~1900 individual TF-target relations. For each set of relations, we tested if the TF correlates better with its target genes than with non-target background genes (Methods, Supplementary Fig. 7). When using the TF protein expression for the comparisons, we found that 100% (10/10 sets) correlated significantly better with its targets than with non-target genes. When the same comparison was made using TF measurements at the RNA level, only 60% correlated better with the targets. When considering only cells in steady-state conditions, the corresponding percentages were 50% when transcription factors were measured at the protein level and 10% when they were measured at the RNA level (Supplementary Table 2). In conclusion, we find that transcription factors predict expression of their target genes significantly better when measured at the protein level than at the RNA level, even when thousands of individual TF-target relations are considered.

We then decided to focus our analysis on POU5F1 as it is an essential stem cell factor that has both positive and negative regulatory functions and is turned off upon differentiation. We first investigated the relationship between either POU5F1$_{RNA}$ or POU5F1$_{protein}$ expression levels and the mRNA expression levels of a subset ($n = 8$) of downstream POU5F1 trans-regulatory target candidates in ESCs (Methods) (Supplementary Table 3). We analyzed both cells at steady-state (0 h) and cells at the onset of differentiation, including all time points analyzed (0 h, 24 h, 48 h). Overall, the Pearson correlation and the regulatory link weights of POU5F1$_{protein}$ expression to POU5F target genes are stronger than those for POU5F1$_{RNA}$ expression (Fig. 3a, Supplementary Table 3).

Given the strong correlation between POU5F1$_{protein}$ and candidate target$_{RNA}$, we then investigated the possibility to use regulatory link weights between other TF$_{protein}$ and candidate target$_{RNA}$ expression as evidence that this transcription factor directly regulates the expression of the corresponding genes. To test this, we ranked all TF-target link weights (Methods) of either POU5F1$_{protein}$ or POU5F1$_{RNA}$ as TF to all expressed genes in a set of samples. Next, we assigned all eight initial POU5F1-target pairs described above as positive targets and all the other genes as negative targets ($n = 8602$). We used this classified ranked list to create a receiver operating characteristic (ROC) curve to identify the cost of identifying negative targets, i.e., the false positive rate

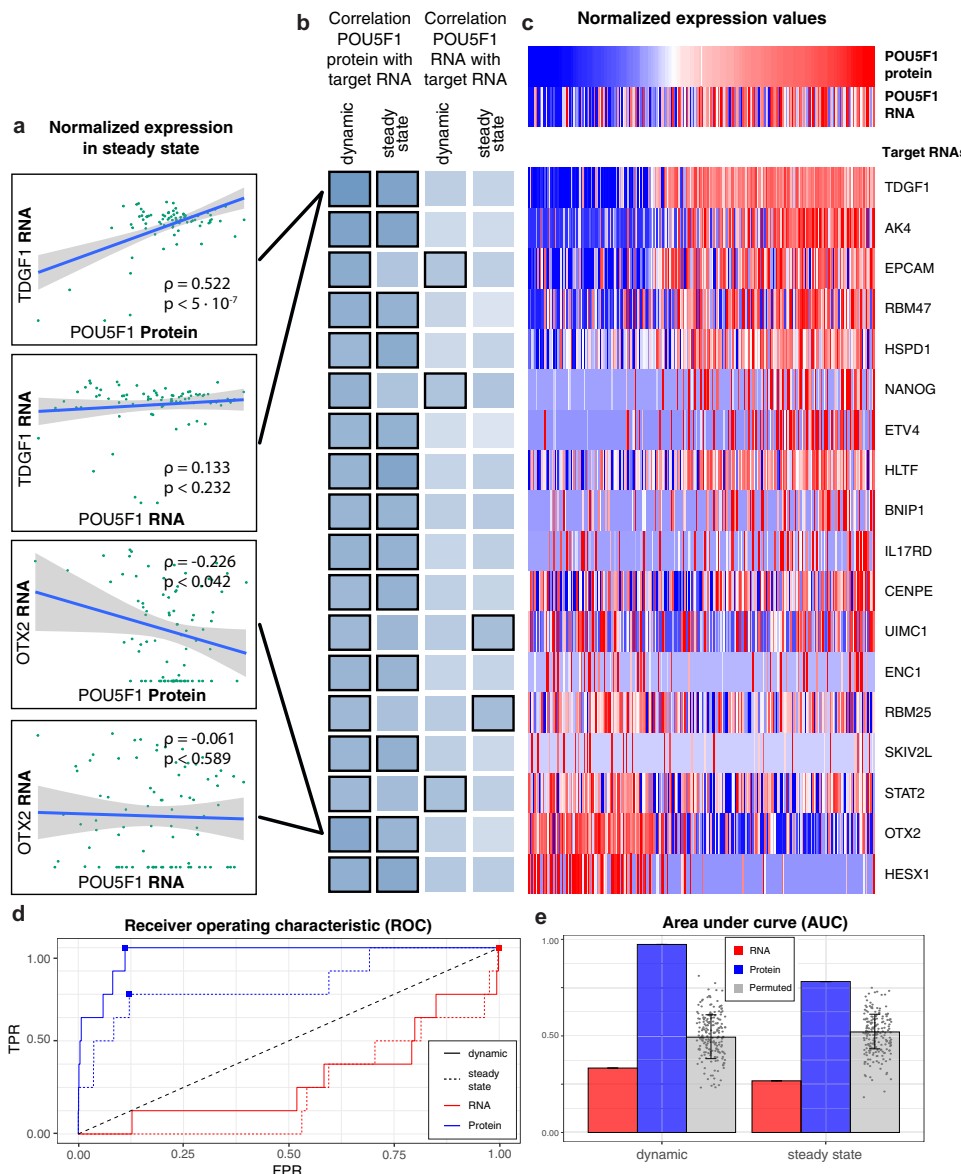

**Fig. 3 Transcription factor protein but not mRNA correlates with target mRNA. a** Scatter plot of POU5F1 expression in 0 h single cells (steady state), both at the level of RNA and protein compared to the RNA expression of POU5F1 regulatory target genes TDGF1 (upper panels) and OTX2 (lower panels). Blue line represents linear model between the two expression patterns. **b** Correlation between POU5F1 protein and RNA levels with the RNA expression of POU5F1 regulatory targets in steady state (0 h) or dynamic (0, 24, and 48 h) conditions. Darker shading indicates stronger correlation. Bold outlining indicates in which of the analysis groups the correlation was considered to be significant (FDR < 0.05). **c** Heatmap with expression levels of POU5F1 at the protein and RNA level and the RNA expression of POU5F1 regulatory target candidate genes. The color scale relates to high (red) and low (blue) expression. The columns in the heatmap are ordered based on the protein expression of POU5F1. **d** Red lines represent POU5F1 (protein)—target (RNA) ROC curves and blue lines represent POU5F1 (RNA)—target (RNA) ROC curves based on regulatory link weights. Solid line represents dynamic conditions, dotted lines indicate steady-state conditions. Filled squares represent the Youden index of the different ROC curves. **e** Quantification of total area under the curve for ROC (see Fig. 3d) including a permuted control of 200 permutations each AUC represented as a dot.

(FPR), based on the number of positive targets, i.e., true positive rate (TPR), identified. We did this using four different analysis groups where we calculated the correlation between $POU5F1_{protein}$ or $POU5F1_{RNA}$ and $target_{RNA}$, and included either only steady-state (0 h) cells or all cells (0, 24, 48 h) (Fig. 3d, e).

To estimate the power of the different correlation analysis groups, we calculated the area under the curve (AUC) of the ROC curves, resulting in the following correlations: $POU5F1_{protein}$ and $target_{RNA}$ with steady-state (AUC = 0.80) or all cells (AUC = 0.97), correlation of $POU5F1_{RNA}$ and $target_{RNA}$ with steady-state (AUC = 0.23) or all cells (AUC = 0.29) (Fig. 3e). To test what values of AUC we would expect by chance, we performed a

permutation test ($n = 200$) with randomization of which genes that are the true targets (AUC = 0.50 ± 0.092) (Fig. 3e). As expected, the AUC results show that using data from cells undergoing a dynamic change gives better power to detect positive POU5F1 targets than only using data from steady-state, and that this is true for correlation to either $POU5F1_{protein}$ or $POU5F1_{RNA}$. Importantly, the results also show that $POU5F1_{protein}$ expression levels better predict regulatory targets than $POU5F1_{RNA}$ levels.

We next asked if we could identify POU5F1 regulatory targets that were not in our known candidate list. We did this by selecting the top five percent correlated POU5F1 target pairs in

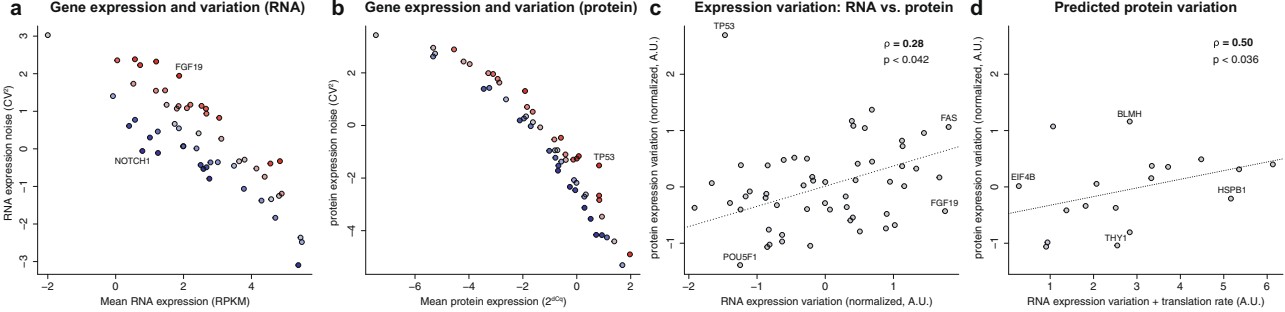

**Fig. 4 RNA variation is not predictive for protein variation.** Patterns of gene expression variation at steady-state in S-phase cells. **a** Relation between mean RNA expression mean (RPKM) and RNA expression variation ($CV^2$). Colors indicate variably (red) or stably (blue) expressed genes. **b** Similar to **a**, but for protein. **c** Normalized gene expression variation (residuals) at the RNA and protein levels correlate moderately (Spearman's rho = 0.28). Genes of interest are indicated. **d** A simple additive model of RNA variation and estimated translation rate effectively predicts protein expression variation (Spearman's rho = 0.50). A subset of genes from Fig. 3c are included here, as public data were not available to estimate translation rate for all genes of interest.

data from cells at steady-state (0 h) or cells from all time points analyzed (0 h, 24 h, 48 h). We further filtered the list of gene-pairs by requiring, first, evidence that POU5F1 binds in the vicinity of the transcription start site of the identified target gene and, second, that the gene-pair correlates significantly in at least two analysis groups (Fig. 3b). Gene enrichment analysis using the Reactome pathway database of the resulting 18 predicted POU5F1 targets identified pathways that are directly related to the regulation of human pluripotent stem cells, specifically *Transcriptional regulation of pluripotent stem cells* (*p*-value = 0.0006), and *POU5F1, SOX2, NANOG activate genes related to proliferation* (*p*-value = 0.0006). Also, POU5F1 was identified as the most significant transcription factor candidate in a query of the TRRUST Transcription Factors 2019 data set (*p*-value = 0.002)[22]. In summary, our analysis highlights the power of using protein level measurements to identify transcription factor regulatory targets in single cell expression data, and also introduces an approach that provides orthogonal evidence to that from ChIP-seq for identifying target genes for transcription factors.

**Gene expression variation.** A major strength of single-cell gene expression profiling is the possibility to study how gene expression varies between cells. Importantly, the factors that impact variation of expression at the mRNA or protein level differ. To date, little is known about the extent to which RNA expression variation translates into protein expression variation. To analyze differences in expression variation at the RNA and protein level, we focused on cells harvested at the 0 h time point that were computationally assigned to S-phase thereby excluding confounding effects of differentiation and cell cycle. Similar results were obtained for the cells in G2/M-phase and—to a lesser extent—the entire cell population (Supplementary Fig. 8) highlighting that these patterns are not S-phase specific.

We calculated the coefficient of variation of each gene at the mRNA and protein level, respectively. For mRNA, it is well understood that gene expression variation depends on the mean expression[23] (Fig. 4a). We demonstrate a similar dependence for proteins (Fig. 4b). This dependence reflects not only biological variability but also technical components such as sampling noise. We subsequently normalized the mRNA and protein expression variation data to their respective mean expression to obtain independent measurements of mRNA and protein expression variation (see Methods). We find that there is generally a weak correlation between the variation of expression at the mRNA and protein level expression (Spearman's rho = 0.28, N = 55, Fig. 4c).

For some genes, there is good agreement, such as the pluripotent factor POU5F1 which is stably expressed both at the levels of RNA and protein, or FAS which shows both substantial RNA and protein expression variability. Other subsets of genes, however, are more variable at the mRNA level (e.g., FGF19), or at the protein level (e.g., TP53). These results highlight that variation at the mRNA level is not generally propagated to protein, and therefore RNA and protein levels should both be considered independently when assessing the impact of expression variation on gene function. This finding has important implications for single-cell gene expression studies where gene sets for further analysis are often selected based on expression variability.

RNA and protein expression variation are complex, co-dependent processes. In order to investigate factors that determine differences between mRNA and protein expression variation, we considered the translation of mRNA to protein. On the basis of reporter assays, it has been suggested that highly translated genes show a higher variability of protein levels[24]. Here, we estimated translation rates (see Methods) and present additional evidence supporting this hypothesis for dozens of genes under physiological conditions as we observe a positive correlation between the estimated translation rates and protein expression variabilities (Spearman's rho = 0.49, N = 18). Interestingly, a simple addition of RNA expression variability and estimated translation rate yields an even better estimate of protein expression variability (Spearman's rho = 0.50, Fig. 4d). This finding supports the notion that protein variability in single cells can be decomposed into RNA variability and noise originating from translation, and for the first time provides evidence for it to be physiological and generalizable.

## Discussion

The more direct roles of proteins in maintaining cellular functions compared to transcripts, together with the recognized importance of post-translational regulation, render single cell protein analysis an important complement of comprehensive RNA analyses of cell state and to decipher gene regulation circuitry.

Overall, we observe that the observed patterns of mRNA and protein co-expression reflect the snap-shot dynamics of single cell gene expression measurements with an inherent temporal element of delay between transcription and translation[16], as well as known characteristics of mRNA and protein molecules, including their respective expression kinetics[25], half-lives, and copy numbers[2]. Results from single-cell mass spectrometry[26] support these results. We note that our measurements are not corrected

for technical measurement noise, and that incorporating reliability estimates for RNA and protein measurements may provide more acute view of mRNA–protein relationships[27].

We observed relationships between mRNA and protein variability that are consistent with earlier observations where cells employ mechanisms to either reduce or amplify intra- and inter-cell protein expression variability introduced by burst-like/noisy mRNA transcription[25]. While protein globally exhibits lower variation than RNA, intrinsic RNA variability of individual genes is not predictive of their protein variability. As shown before, translation efficiency explains part of this discrepency[24]. Additionally, protein stability can play a role in the relationship between extrinsic mRNA and protein variability[28]. Short-lived proteins can track fluctuating mRNA levels closely, while levels of protein that degrade slowly fail to follow the rapid fluctuations of mRNA.

The SPARC procedure described herein can help resolve regulatory networks or monitor developmental processes and cellular responses to e.g., genetic or chemical perturbations. In this work, we showed that transcription factors, when measured at the protein level, correlate well with the RNA expression of their targets, but there is little correlation when the transcription factors are measured at the RNA level. We propose that the approach can be used for the analysis of transcript isoform usage and protein expression as explored in Genshaft et al.[31], and will be valuable for analyses of the relationship between mRNA and protein expression variation in general. The proximity-based protein assays are well suited to detect post-translational modifications[32], add detection specificity, and can be scaled up via a sequencing readout[33], allowing measurement of even greater numbers of proteins and protein modifications in individual cells.

Since SPARC employs the Smart-seq2 protocol, future studies could utilize the sizeable fraction of intronic reads to further dissect gene regulation and expression dynamics. For example, we hypothesize that RNA and protein co-profiling can be used to predict future cell states similar to RNA velocity that employs spliced and un-spliced transcripts[29,30]. Updated versions of the Smart-seq2 protocol, such as Smart-seq 2.5 and Smart-seq3, furthermore allow for greater sensitivity and better quantification through the use of UMIs. Finally, random hexamers or specific sequences of interest can be conjugated to magnetic beads in order to extend capture beyond non-polyadenylated transcripts.

SPARC provides sensitive and precise transcript quantification while enabling highly specific and scalable detection of intracellular proteins without the need for invasive cell fixation, in single cells. The approach is built upon two well established methods whose performance have repeatedly been demonstrated. At present, no comparative, orthogonal approaches exist but we foresee great value in the development of complimentary, combined mRNA–intracellular protein high-throughput methods in order to further study the roles of transcripts and proteins in maintaining cellular states and functions.

## Methods

**Cell culture and neural induction of human embryonic stem cells**. Human embryonic stem cell line HS181 (hPSCreg Kle001-A; with consent from donor and with ethical permission from Karolinksa Institutet) was maintained on vitronectin (VTN-XL; Stem Cell Technologies, 07180) in Essential-E8 Medium (ThermoFisher Scientific, A1517001). Cells were passaged as clumps with gentle cell dissociation reagent (GCDR; Stem Cell Technologies, 07174) when necessary. Two days before neural induction, cells were harvested with GCDR and plated in Essential-E8 supplemented with 10 uM Rho-kinase inhibitor Y27632 (Stem Cell Technologies, 72304) on VTN coated 6-well culture dish to yield approximately 80% confluence the next day. When cells reached 100% confluence, neural differentiation was induced (Day 0/0 h) using dual smad inhibition (Maroof et al., 2013). Briefly, cells were washed with 1x DPBS and neural induction medium (NIM) was added, consisting of KnockoutTM DMEM (10829018), 15% KnockOutTM Serum Replacement (A3181501), 1x GlutaMax (35050038), 1x non-essential amino acids

(11140035), 1% penicillin/streptomycin (1514022) (all ThermoFisher Scientific), supplemented with 2 uM tankyrase inhibitor XAV939 (Sigma-Aldrich, X3004), 100 nM ALK2/3 inhibitor LDN193189 (Miltenyi Biotech, 130-106-540), and 10 uM ALK4/5/7 inhibitor SB431542 (Millipore, 616464). Medium was replaced on days 1 and 2.

To follow early neural differentiation, cells were harvested at indicated time points (0 h; 24 h; 48 h). Cells were washed twice with DPBS to remove floating dead cells, and subsequently treated with Accutase (Sigma-Aldrich) for 5–15 min until a single cell suspension was obtained. Cells were collected in 13 ml DPBS and then centrifuged at $300 \times g$ for 5 min, washed with DPBS, centrifuged again, and resuspended in DPBS. Cells were kept on ice until single cell sorting.

**Single cell isolation**. The single cells were incubated on ice with a viability dye (LIVE/DEAD Viability Kit, Molecular Probes, L3224) for approximately 15 min and passed through a 40 μM filter before sorting. Cells were sorted on a BD FACS ARIAIII into a 96-well plate. We used a gating strategy to exclude off debris, doublets, and cells positive for EthD-1, a nucleic acid dye that enters cells with damaged membranes. After cell sorting, the plate was centrifuged at 700 g and 4 °C for 1 min and then quickly transferred to dry ice. The plates were stored at −80 °C until processed.

For the SPARC protocol, cells were sorted into 1.5 μl TE buffer (pH 8.0 Invitrogen, AM9858) with the following components: 1% NP-40 (Thermo Fisher, 28324), 0.1% Triton X-100 (Thermo Fisher, 28314), 0.1% Sulfobetaine (Sigma-Aldrich, 82804-50 G), 150 mM NaCl (Ambion, AM9760G), 10 mg/ml BSA (Ambion, AM2616), 2 U SUPERase In RNase Inhibitor (Ambion, AM2696), 1X HALT protease inhibitor (Thermo Fisher, 78430), and 1:1,250,000 ERCC (Invitrogen, 4456740).

For the Smart-seq2 protocol, cells were sorted into 4 μl TE buffer (pH 8.0) with the following components: 0.1% Triton X-100 (Thermo Fisher, 28314), 1 U SUPERase In RNase Inhibitor (Ambion, AM2696), 1:4000000 ERCC spike-in (Ambion, 4456740), 2.5 mM dNTPs (Thermo Fisher, R0191), and 2.5 μM oligo-dT (Integrated DNA Technologies IDT, 5′-/5BiotinTEG/AAGCAGTGGTATCAACG CAGAGTA CT₃₀VN-3′, where V is either A, C or G, and N is any base). For multiplex protein detection only, cells were sorted into 1.5 μl TE buffer pH 8.0 with the following components: 1% NP-40 (Thermo Fisher), 0.1% Triton X-100 (Thermo Fisher), 0.1% Sulfobetaine (Sigma- Aldrich, KEYH14DECD93-25G), 150 mM NaCl, 10 mg/ml BSA (Ambion, 4456740), and 1X HALT Protease Inhibitor (Thermo Fisher).

On each 96-well sorting plate, we included: (i) population controls of 100 sorted cells/well in duplicate; (ii) buffer, no-cell control in triplicate; (iii) 100 cell equivalent lysate prepared from the SK-MEL-30 cell line. The SK-MEL-30 cell lysate was prepared in bulk, aliquoted, and added to every plate in triplicate as an inter-plate control.

**Oligo-dT bead preparation**. For every reaction, 5 μl Dynabeads MyOne Streptavidin T1 (Invitrogen, 65602) were washed twice with 1.45 μl washing solution containing 100 mM NaOH (Sigma-Aldrich, S8045-500G), 50 mM NaCl (Ambion, AM9760G), and UltraPure DNase/RNase-Free Distilled Water (Invitrogen, 10977035). The beads were then washed with 1.45 μl RNAse-free bind and wash solution containing 0.01 mM Tris (Invitrogen, AM9855G), 1 mM EDTA (AM9260G), 2 M NaCl (Ambion, AM9760G), and UltraPure DNase/RNase-Free Distilled Water (Invitrogen, 10977035). The beads were then mixed with 2 μl RNAse-free bind and wash solution and 0.1 μl oligo-dT (IDT, 5′-/5BiotinTEG/ AAGCAGTGGTATCAACGCAGAGTA CT₃₀VN-3′) and incubated in ambient temperature for 15 min. The beads were finally stored in 1 μl 1% BSA (Ambion, AM2616) in TE buffer (Invitrogen, AM9858) and incubated on a rotator overnight at 8 °C. Before use, the buffer was exchanged with RNA and Protein lysis buffer.

**mRNA capture, reverse transcription, and pre-amplification**. Lysis plates with FACS sorted cells were centrifuged at $700 \times g$ for 10 s and thawed on ice. Smart-seq2 reference samples were incubated at 72 °C for 3 min and directly placed back on ice. SPARC samples were pipette-mixed while 1 μl of prepared beads was added to each reaction. The SPARC plate was then incubated for 10 min with orbital shaking at 1000 rpm. Plates were then centrifuged at $700 \times g$ for 10 s and placed on a magnetic rack (Alpaqua Magnum FLX) where 1.7 μl from each well was transferred for protein analysis. The Smart-seq2 and SPARC samples were supplied with 6 μl and 10 μl reverse transcription mix, respectively. The mixes contained: 100 U SuperScript II reverse transcriptase, 1x First Strand Buffer, 5 mM DTT (all Invitrogen, 18064014), 10 U SUPERase In RNase Inhibitor (Invitrogen, AM2696), 1 M Betaine (Sigma-Aldrich, 61962-250 G), 6 mM MgCl₂ (Invitrogen, AM9530G), 1 mM of each dNTP's (ThermoScientific, R0192), 1 μM TSO (5′-AAGCAGTGG-TATCAACGCAGAGTACATrGrG+G-3′, Exiqon as described in Picelli et al., 2014 and UltraPure DNase/RNase-Free Distilled Water (Invitrogen, 10977035). The Smart-seq2 samples were centrifuged for 700 g for 10 s and then placed in a thermal cycler. The SPARC samples were pipette-mixed prior to incubation in a thermal cycler. Both samples sets were processed using the following program: 42 ° C for 90 min, 10 cycles with 50 °C for 2 min and 42 °C for 2 min and finally 70 °C for 15 min before holding at 4 °C.

Each first strand cDNA reaction was supplied with 15 μl of PCR mix containing 1x KAPA HiFi HotStart Ready Mix (Kapa Biosystems, KK2601), 1 μM IS PCR primer (5′-AAGCAGTGGTATCAACGCAGAGT-3′, IDT, as described in Picelli et al., 2014 and UltraPure DNase/RNase-Free Distilled Water (Invitrogen, 10977035). The RNA samples were vortexed and centrifuged ($700 \times g$ for 10 s) while the combined RNA/Protein samples were pipette-mixed, prior to incubation in PCR program: 98 °C for 3 min, cycling of 98 °C for 20 s, 67 °C for 15 s and 72 °C for 6 min. Lastly, the final extension was at 72 °C for 30 s prior to holding at 4 °C. Single cells were subjected to 20 PCR cycles while bulk samples (100 cells) were subjected to 14 PCR cycles. Samples were finally purified with AMPure XP beads (Beckman Coulter, A63880) using 0.8X bead to sample ratio.

**scProtein expression analysis**. Cell lysate containing the protein supernatant were transferred to a new 96-well PCR-plate and processed immediately. To each sample, we added 2.1 μl Incubation Solution (Olink Proteomics), 0.3 μl Incubation Stabilizer (Olink Proteomics), 0.3 μl of each PEA A- and B-probe mix (final concentration 100 pM; Olink Proteomics). The probes targeted 92 cellular proteins and 4 controls. The controls included spiked-in GFP, PE, an extension control, and a detection control[15]. Each 96-well plate included a lysis buffer only negative control in triplicate. PCR-plates were briefly vortexed, centrifuged, sealed, and incubated overnight at 8 °C. Following overnight incubation, plates were brought to room temperature, briefly spun down, and 96 μl Extension mix was added to each well. The extension mix contained 10 μl PEA Solution (Olink Proteomics), 0.5 μl PEA Enzyme (Olink Proteomics), 0.2 μl PCR Polymerase (Olink Proteomics), and 85.3 μl UltraPure DNase/RNase-Free Distilled Water (Invitrogen, 10977035). Plates were sealed, gently vortexed, centrifuged and within 5 min of adding the Extension mix, placed in a thermal cycler for the extended reaction (50 °C, 20 min), and pre-amplification of extended PEA probes via universal primers (95 °C, 5 min; 95 °C, 30 s; 54 °C, 1 min; and 60 °C, 1 min) x 17.

The pre-amplified extended PEA products were decoded and quantified using a Fluidigm 96.96 Dynamic Array Integrated Fluidic Circuit on a Biomark HD system. 96 primers pairs (5 μl of each) targeting each PEA probe pair were loaded in the left inlets of the array. A Detection mix containing 5 μl Detection Solution, 0.071 μl Detection Enzyme, 0.028 μl PCR Polymerase (all Olink Proteomics), and 2.1 μl UltraPure DNase/RNase-Free Distilled Water (Invitrogen, 10977035) was added in the right inlets of the array. The 96.96 IFC chip was primed in Fluidigm's IFC HX according to manufacturer's instructions and then run on the Biomark HD system with the following settings: Gene Expression application, ROX passive reference, single-probe assay with FAM-MGB probe. The thermal protocol included thermal mix (50 °C, 120 s; 70 °C, 1,800 s; 25 °C, 600 s), hot start (95 °C, 300 s), and PCR cycling for 40 cycles (95 °C, 15 s; 60 °C, 60 s).

**Library preparation and sequencing**. Purified cDNA (75 ng) was used as input to the Nextera XT DNA library preparation kit (Illumina, FC-131-1096), following the manufacturers protocol with the modification of using 1:5 of reagent volumes. Indexing primers (Illumina, FC-131-2001; FC-131-2002; FC-131-2003; FC-131-2004) were diluted 1:2 in UltraPure DNase/RNase-Free Distilled Water (Invitrogen, 10977035) prior to use. Samples were finally pooled and purified with AMPure XP beads (Beckman Coulter, A63880) using 0.6x bead to sample ratio and concentrated by eluting in 60% of the corresponding input sample volume using Elution buffer (Qiagen, 19086). The final sequencing library was quantified using BioAnalyzer High Sensitivity DNA kit (Agilent, 5067-4626) using a region table spanning 100 bp to 1000 bp. The pooled library was sequenced on two lanes of Illumina HiSeq2500 using single read 50 bp read length, v4 chemistry. The sequencing was performed at the SNP&SEQ Technology Platform, Science for Life Laboratory, Uppsala, Sweden.

**scRNAseq dataset processing**. Reads were mapped to the human genome (GRCh38) including the sequence of the spike-in RNAs. FeatureCount summarized over annotated genes from the GRCh38.77 version of the human genome including the spike-ins were used to get counts for all exons of annotated genes in the human genome. Samples with less than 10,000 reads mapping to the exons or the fraction of spike-in RNAs were greater than 20% were removed from further analysis. RPM values and RPKM values were calculated for all genes in all samples.

**scProtein dataset processing**. Following the completion of the qPCR run on the Fluidigm Biomark HD system, we visually inspected the amplification curves. Samples showing evidence of failed or poor amplification reactions were excluded from further analysis. Next, the raw Cq data (log 2 scale) from the Fluidigm Biomark HD system was exported and processed. First, samples were excluded if no signal was detected in any of control assays (extension control, incubation control, or detection control), or if the signal in any of the controls was greater or less than 2 standard deviations (SD) of the mean value across all samples measured on a 96.96 IFC Biomark chip. Next, the remaining Cq values were normalized for intra-plate variation with the extension control ($Cq_{assay} - Cq_{ExtCtl}$) yielding dCq values. Then, for each assay, the dCq values were subtracted from the negative control computed as the lysis buffer mean + 2 × SD. This ensures that observed signals for each assay in the presence of a cell are at least 2 SDs away from any signals observed in the absence of any antigen. Resulting values below zero were set

to zero and the signal was deemed undetected. The cumulative protein sum was calculated by summing across all proteins measured ($n = 92$) per cell.

**Method comparison for RNA analysis**. Samples from both the SPARC protocol and standard Smart-seq2 protocol at 0 h were used to compare similarities and differences between the two protocols. Only genes with log RPKM greater than 1 were used for further analysis. Genes were separated into different biotypes according to GRCh38.77 annotation and number of detected genes per biotype were compared between the two protocols. Read counts for all exons and introns in all genes were counted and the distribution across the genes were compared between the two protocols. Logarithmic mean expression (LME) for each gene was calculated for all samples irrespective of protocol, and for each protocol by itself LMESPARC-seq and LMESmart-seq2. Differential expression between the protocols (DE-prot) per gene were calculated by dividing the LMESPARC-seq with the LMESmart-seq2 value per gene. Both the LME and the DE-prot were taken into account by multiplying the two values to identify the differences between the two protocols. Genes where the product of the two was greater than 8 was considered to be different and analyzed for differences in lengths and gene biotype.

**Cell cycle assignment**. RNA samples were normalized using the Seurat 2[24] package and the samples were scaled by the number of detected genes per sample. Scores for each sample being in either S phase or G2/M phase were calculated using the Seurat 2 package and at the same time predicted to belong to either the G1, S, or G2/M phase.

To further explore the relationship between cell cycle phase and protein expression, hESCs were labeled with the Live cell DNA dye Vybrant DyeCycle Violet (Invitrogen, V35003) and sorted by cell cycle phase (G1, S or G2/M) in triplicate at 100 cells per well. Cells were sorted in the following lysis buffer: 2 μl TE buffer (pH 8.0) with the following components: 1% NP-40 (Thermo Fisher, 28324), 0.1% Triton X-100 (Thermo Fisher, 28314), 0.1% Sulfobetaine (Sigma-Aldrich), 150 mM NaCl (Thermo Fisher, AM9760G), 10 mg/ml BSA (Thermo Fisher, AM2618), and 1X HALT Protease Inhibitor (Thermo Fisher, 78430). Cells were then processed for multiplex PEA analysis as described above.

**PCA analysis and pseudotime analysis**. The normalized and scaled data from the cell cycle prediction were further scaled by the scores for the S phase and the G2M scores to remove the cell cycle dependency of the samples using the Seurat package. Dimensional reduction analysis using tSNE were used to reduce the dimensionality of the sample data. The three dimensions from the tSNE analysis was then used by SCORPIUS[19] to predict a linear pseudo time through all the samples with a score between 0 and 1. Average pseudotime scores for the 0 h samples and 48 h samples were calculated. If the average pseudotime score for the 0 h samples was higher than the average 48 h samples the pseudotime was re-calculated by subtracting the pseudotime with 1 and then multiplying by −1 to change the order of the samples and maintaining the pseudotime distances between samples. Pseudotime scores were then multiplied with 48 to reflect a pseudotime over 48 h. Genes that change over time were identified by SCORPIUS (adjusted p-value <0.05) and separated into different modules dependent on expression pattern over time.

**Comparison of RNA and protein changes over time**. To test whether the expression levels for both RNA and protein level show the same directional changes, we applied a linear model for expression over time for both the RNA logged RPKM values ($lRPKM(pt) = klRPKM \times pt + IlRPKM$) and for protein Cq values ($Cq(pt) = kCq \times pt + ICq$). We then tested if the slope of the RNA levels for a gene could predict the slope of the protein for the same gene with the linear model $kCq$ ($klRPKM$) = x $klRPKM$ + z. We fitted it and got the linear model $kCq$ ($klRPKM$) = 0.49 $klRPKM$ + 0.01.

**Gene regulatory network analysis—selection of POU5F1 targets**. Initial sets of potential target for TFs were downloaded from enrichr databases. The three TF target enrichment databases used were ENCODE_and_ChEA_Consensus_TFs_from_ChIP-X library as a mean to identify genes where the TF bind in vicinity, Enrichr_Submissions_TF-Gene_Coocurrence library to identify genes that are co-expressed with TFs, and TRRUST_Transcription_Factors_for known TF targets according to literature. In our analysis we considered TRRUST targets identified in mouse also as true targets.

To reduce the number of false-positive targets based on the approaches above and identify genes that are trans-regulated by POU5F1, an initial set of potential POU5F1 targets were selected based on two criteria: (1) identified as one of the genes that changed over pseudotime using SCORPIUS (adj p-value <0.05); and (2) reported as bound by POU5F1 in vicinity of the TSS by the curated set of ChIP-seq data described in the section below. The intersection of the two assumptions gave a subset of 8 high confidence target genes (AK4, DHRS3, DUSP6, KLHL4, OTX2, PRR14L, TDGF1, WLS).

A set of genes where POU5F1 binds in the vicinity of the TSS in primed hESCs were identified using ChIP-seq data[35]. An initial set of 11 hESC ChIP-seq experiment data (SRX017276, SRX021069, SRX021071, SRX1053369, SRX1053370, SRX1053378, SRX1053379, SRX266859, SRX702065, SRX702066, SRX702069) were collected from CHiP-Atlas[34] by filtering with POU5F1 as TF and a distance

from TSS of 1 kb. For the remaining samples, hierarchical clustering of the Euclidean distance of binding scores across all genes and experiments was performed. A final set of seven experiments (SRX017276, SRX021069, SRX021071, SRX1053378, SRX1053379, SRX702065, SRX702066) that clustered as primed hESC in the hierarchical cluster were kept for further analysis. Genes with a reported binding score in ChIP-Atlas in at least two samples were considered as the curated set of POU5F1 target genes.

Regulatory link weights for TF-target pairs were calculated using GEne Network Inference with Ensemble of trees (GENIE3)[35] between the TF expression pattern and the expression pattern of expressed annotated coding genes in the cell. To remove noise, only genes with a mean RPKM of 2 were considered. To evaluate whether the targets were upregulated or downregulated, i.e., positive or negative correlation, calculation of the Pearson correlation was done.

**Gene expression variation.** Gene expression variation was estimated using the squared coefficient of variation ($CV^2$). As variation and mean expression are inherently linked due to sampling properties, we perform a polynomial fit of variation to mean expression in log-space. We then considered the residuals of the individual gene measurements with regard to this fit as mean-independent gene expression variation measurements (termed "normalized variation"). To avoid in-silico biases, data was not computationally de-noised using batch-effect removal tools. Instead, cells were strictly filtered and quality controlled. Specifically, outliers were removed using pseudo-time estimations from SCORPIUS, and only S-phase cells were considered, using the cyclone package of SCRAN. Mass-spectrometry data for the estimation of the translation rate for hESCs (E14 cells[36]) was retrieved from FunCoup PaxDB[37] statistics. Translation rates are approximated by the log2 ratio of protein abundances vs. RNA RPKM measurements. Due to the different dynamic ranges of the RNA variability and translation measures, the "combined normalized variation and translation rate" reflects the sum of normalized RNA expression variation and 0.75 times the estimated translation rate. Lines in Fig. 4c, d show linear least total square fits.

**TaqMan gene expression analysis.** Total RNA was isolated from the HS181 cell line using the miRNeasy Micro Kit (Qiagen, 217084). We used the One-Step RT-PCR System (Thermo Fisher Scientific, 12574026) with the Taqman Assays IKBKG (Hs00415849_m1, 4453320), METAP1D (Hs00994998_m1, 4448892), HMOX1 (Hs01110250_m1, 4453320), and EIF4B (Hs00973573_m1, 4331182) (all Thermo Fisher). Gene expression was quantified from the isolated RNA at 100 ng/reaction using the Quantstudio real-time qPCR instrument.

**Reporting summary.** Further information on research design is available in the Nature Research Reporting Summary linked to this article.

## Data availability

The SPARC single-cell sequencing data have been deposited at ENA with the project number PRJEB33157. The SPARC single-cell protein data have been deposited at SciLifeLab Data Repository with https://doi.org/10.17044/scilifelab.14207462. The data is reported as described in methods under scProtein dataset processing but are not normalized for cumulative protein sum. Assays are filtered for those where we detected the protein at a level of >3 Cq over background in the 100 cell population control in at least one time point. The processed data to generate all figures, and all supplementary figures and tables are available at GitHub (b97jre/SPARC) and also been deposited at SciLifeLab Data Repository with https://doi.org/10.17044/scilifelab.14207909.

## Code availability

All analysis post mapping and counting has been carried out in R. Package dependencies and code to generate all figures, supplementary figures and tables are available at GitHub (https://github.com/b97jre/SPARC) and also been deposited at SciLifeLab Data Repository with https://doi.org/10.17044/scilifelab.14207909.

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

## Acknowledgements

We thank Ulf Landegren and Lars Feuk for helpful feedback on the manuscript. Sequencing was performed by the SNP&SEQ Technology Platform in Uppsala, Sweden. The facility is part of the National Genomics Infrastructure (NGI) Sweden and Science for Life Laboratory. The SNP&SEQ Platform is also supported by the Swedish Research Council and the Knut and Alice Wallenberg Foundation. C.G. acknowledges funding from the Swedish Research Council (VR) Grant 2017-05229 and support from the Single Cell Proteomics Facility, Science for Life Laboratory, Sweden. M.T. and M.R.F. acknowledge funding from ERC Starting Grant 758397 "miRCell", VR Research Grant 2019-05320 "MioPec", and from the Strategic Research Area (SFO) program of the Swedish Research Council (VR) through Stockholm University. N.D. acknowledges funding from the Swedish Research Council 2015-02424.

## Author contributions

C.J.G., M.D., and M.R.F conceived the concept and analysis framework. M.D., S.B., and S.P. ran the experiments with cell models from J.S. and N.D. J.R. and M.T. designed and performed the data analysis. J.R., M.T., M.R.F., and C.J.G wrote the manuscript. All authors commented on the manuscript.

## Funding

## Competing interests

The authors declare no competing interests.
