## [Peer Review File · Communications Biology]

Reviewers' Comments:

Reviewer #1:

Remarks to the Author:

The submitted publication presents a method for targeted protein and untargeted mRNA readout on single cell resolution that integrates the multiplex Olink proximity extension assay (PEA) with the smart seq 2 protocol. The method was applied on the two day differentiation of ESCs towards neural progenitor cells. Using the single-cell protein and mRNA readout they were able to investigate how mRNA and protein variation correlate over time and how this process is regulated.

Overall the manuscript contains very interesting data and approaches worth publishing the nature com biol. The unstructured, the verbose writing style in some parts, and scientific inaccuracy in other parts makes it hard to judge and follow. Therefore, the current version of the manuscript is premature.

The following are only some examples of the unstructured manuscript

- Some paragraphs of the manuscript contain only references to SI figures
- SI is highly unstructured and needs different presentation forms
- Figure 3 references are not systematic and the text jumps between effects. Figure 3c is not explained.
- Effects linked to the distribution of the cq values are discussed in the second and last paragraphs
- Biological problems are hidden in technical presentations. Example: Page 3, 1 Paragraph the comparison data between sparc and smartseq2 is presented in one bar chart, why adding the differentiation data here?
- Protein target sets are not described (not in SI nor main text) also otherwise stated. Text states that there are more than 90 proteins measured but SI contains only 62. Info hidden in material and methods.
- Figure 2 spans over three separate separated paragraphs
- Fig S7 and S8 are mixed up
- And much more...

One suggestion is to shorten the manuscript and focus on the elementary findings.

The main scientific critiques are

- When characterizing the neuronal differentiation from pluripotent stem cells I would expect the main marker profiles and time courses and protein and mRNA level and not only 2TF
- The selected proteins seem to be designed for something else. It would be wiser to choose fewer targets with more clear biological question even if this is only a proof of principle experimental set up.
- Sampling time for the velocity analysis in a proof of principle for new technology is wrongly chosen (realized by the authors in the text).
- Velocity vectors going unexpectedly in both directions; it is uncertain if the analysis is fully correct in Figure 2. They explain the arrows from 24h to 0h with 24h being a "transition state". But how do the cells come to 24h?
- Oct 4 has a half-life time from about 90 min an offset between mRNA and protein level with a sampling time of 24 h cannot be expected.
- Why focusing so much on Oct4? For the neuronal development more markers are in foreground. Are those not working?
- It is not surprising that the network target prediction is working better for Oct4 on the protein level than mRNA since there shouldn't be much mRNA transcripts for a TF so the noise and technical variances between single cells is larger.
- Probably very variable amplification biases between the RNA detection (smartseq) and the protein detection (a qPCR)? A synonymous readout would produce much better/ easier to handle datasets
- Unclear how the protein level normalization has been done. If the data has been normalized per single cell most of the biological variation has been averaged? Indication for this is that cq values are very similar for all proteins.
- Is the reason of low variability of the protein levels due to insensitivity of the qPCR method?
- Cite seq data has a much higher single cell variability, why? Could this indicate that a the

normalized qPCR is the problem. If so the entire results to figure 4 is affected.

- Is the low cell number a problem for all analyses?
- It is fine to realize that protein and mRNA levels are not redundant but the technical proof for this is too low. The protein method must be validated with an alternative method like Cite seq for surface proteins.

Minor:

- Do they use the whole mRNA count for the RNA-protein velocity or only the unspliced mRNA count?
- Keyword misleading. There is microfluidics, expect a commercially available platform.
- No info about the antibodies and design of the antibody probes – Sequences, washing steps? Unspecific bindings? Unbound antibodies will also hybridize and give signal.
- These problems are mentioned in Supplemental Results and Figures but kind of glossed over otherwise.
- Fig 1 - A + B Very simplified graphics, almost no info about the antibody PEA method
- Fig 4 -Central findings (variability mRNA/protein) are somehow presented very underwhelmingly with very few examples named – why not choose a plot that shows variation for each of their calculated values?
- Fig. 1c shows only 4 proteins are shown, Figure S2, and Table S1 summarizes

Reviewer #2:

Remarks to the Author:

The authors report single-cell protein and RNA co-profiling (SPARC) for simultaneously measuring global mRNA and large sets of intracellular protein in single cells. With SPARC analysis they demonstrated that mRNA and protein measurements in single cells provide different and complementary information regarding cell states. However, major concerns for this study lack the novelty for SPARC which simply combined two well-established Smart-seq2 for RNA analysis and PEA for protein analysis as well as sufficient performance evaluation of SPARC.

In recent years there are many methods reported for simultaneous analysis of protein and RNA (Nat Methods 2016, 13, 269; Nat Biotech 2017, 35, 936; Cell Syst 2018, 7, 398; Cell Syst 2018, 6, 531; Nat Commun 2019, 10, 3544) including one study from author group (Cell Reports 2016, 14, 380-389). Systematic performance comparison with at least one existing method is required to validate SPARC in terms of sensitivity, accuracy, reproducibility, and quantitative dynamic range. Modification of current Smart-seq2 method is not novel with incremental improvement in performance.

Minor comments:

1. In page 3 it is not clear why 89 proteins out of the panel of 96 proteins had detectable levels in single cells, while only 87 proteins in the 100 cell control. Please explain this.
2. In page 11, significant protein loss may occur due to substantial surface absorption when cell lysate containing the protein supernatant is transferred to a new 96-well PCR-plate. Evaluation of protein recovery at the single-cell level for this transfer is needed for precise quantification.

Reviewer #3:

Remarks to the Author:

The authors measured the abundances of mRNAs and proteins in single cells as they undergo differentiation, and used the data to explore the covariation of transcripts and proteins. They also examined the ability to identify the target genes of TFs based on the correlation between TFs and mRNAs and found that protein abundances of TFs are much more predictive of TF target genes. Furthermore, they report that protein expression variation is lower than mRNA variation.

I found the study interesting and the conclusions generally well substantiated. I particularly like the requirement for pairwise protein detection, which ensures higher specificity compared to methods that use a single antibody per protein. I have a few suggestions to further strengthen the

conclusions and their description:

1) It seems to me that the major methodological advance of SPARC over Darmanis, S. et al. (Ref. 11) is replacing the PCR readout with Smart-seq2. I think stating this explicitly will be helpful for clarifying the methodological advance.

2) Measurement noise can have large and decisive influence on interpreting protein - RNA correlations <https://journals.plos.org/ploscompbiol/article?id=10.1371/journal.pcbi.1005535> The authors should incorporate reliabilities in their estimates or at least acknowledge the influence of noise. Note that measurement noise might contribute to the higher correlations for the pseudotime trends since some of the noise is averaged and reduced.

3) The lower variability of the protein levels seems consistent with our observations of lower protein variability in Figure 5 and Figure S3 of <https://www.biorxiv.org/content/10.1101/665307v3>. These observations use different data and analysis and mutually support each other. The author may want to discuss this connection.

4) This description in the introduction is a bit simplistic

"While bulk sample measurements generally report good correlation between mRNA and protein expression, little is known about the extent of correlation at cellular resolution in systems at steady-state or undergoing a dynamic transition."

Some of the referenced studies reported much larger correlations than other referenced studies, and some of these differences are due to considering or not considering measurement noise as well as to computing the correlations across genes or across conditions. The authors may want to be a bit more specific in their description and what is meant by a "good correlation".

5) The paper is clearly written and generally easy to follow. At some places, it can benefit from replacing qualitative statements with more exact quantitative statements, e.g., "large sets of intracellular protein" with "sets of about 100 intracellular proteins".

Generally, I find this paper strong. After incorporating the suggestions listed above, it likely will become one of the strongest papers in the field. SPARC makes very significant improvements over competing methods such as CITE-seq and REAP-seq, and its application demonstrates important results, such as the higher predictive potential of protein measurements for TF-targets.

Nikolai Slavov

Please find attached detailed responses to the reviewers.

Reviewer #1 (Remarks to the Author):

The submitted publication presents a method for targeted protein and untargeted mRNA readout on single cell resolution that integrates the multiplex Olink proximity extension assay (PEA) with the smart seq 2 protocol. The method was applied on the two day differentiation of ESCs towards neural progenitor cells. Using the single-cell protein and mRNA readout they were able to investigate how mRNA and protein variation correlate over time and how this process is regulated.

Overall the manuscript contains very interesting data and approaches worth publishing the nature com biol. The unstructured, the verbose writing style in some parts, and scientific inaccuracy in other parts makes it hard to judge and follow. Therefore, the current version of the manuscript is premature.

The following are only some examples of the unstructured manuscript

- Some paragraphs of the manuscript contain only references to SI figures
- SI is highly unstructured and needs different presentation forms
- Figure 3 references are not systematic and the text jumps between effects. Figure 3c is not explained.
- Effects linked to the distribution of the cq values are discussed in the second and last paragraphs
- Biological problems are hidden in technical presentations. Example: Page 3, 1 Paragraph the comparison data between sparc and smartseq2 is presented in one bar chart, why adding the differentiation data here?
- Protein target sets are not described (not in SI nor main text) also otherwise stated. Text states that there are more than 90 proteins measured but SI contains only 62. Info hidden in material and methods.
- Figure 2 spans over three separate separated paragraphs
- Fig S7 and S8 are mixed up
- And much more...

We implemented several of the suggestions with regard to the structure and clarified various aspects in the main text. We fixed and re-formatted the suppl figures.

The number of proteins varies across different analyses since different filters need to be applied. These filters are described in detail in the Methods section.

Cq variation needs to be touched upon early on in a global manner. Later on, we specifically address gene expression variation on the level of individual genes. We now tie these observations together in the discussion.

One suggestion is to shorten the manuscript and focus on the elementary findings.

The main scientific critiques are

- When characterizing the neuronal differentiation from pluripotent stem cells I would expect the main marker profiles and time courses and protein and mRNA level and not only 2TF

We focused on POU5F1 and SOX2 because they are important and well-studied pluripotency markers. While we indeed drive embryonic stem cells towards neuronal differentiation, the observed time points represent very early stages of this process. We expect, therefore, more drastic and visible changes in pluripotency markers such as POU5F1 and SOX2 as compared to neuronal markers.

We would also like to emphasize that the novelty of this study derives from the newly developed profiling method, SPARC, which overcomes major current limitations to study high-plex mRNA and protein in single cells. The dynamic cellular model system is then used to illustrate the quantitative power of the method.

- The selected proteins seem to be designed for something else. It would be wiser to choose fewer targets with more clear biological question even if this is only a proof of principle experimental set up.

For this proof-of-principle study, proteins were selected to evaluate the value of measuring proteins across a wide range of cellular processes, including pluripotency, cell cycle and metabolism. Since a major advancement of SPARC over other methods such as CITE-seq is the possibility to detect proteins independent of cellular localization, the panel specifically includes many intracellular proteins, including nuclear ones. We specifically avoided focusing on neuronal differentiation as the aim of this study to make more generalizable statements about RNA and protein expression as well as their variation and covariation independent of the specifics of the model system.

Information about the protein panel design has been included in the manuscript. The text is included below for quick reference:

We developed an exploratory multiplex PEA panel for single cell analysis in collaboration with Olink Proteomics, involving 92 proteins and focused on intracellular proteins of interest for our investigation. The panel includes proteins across different functional groups related to, for example, pluripotency, neurogenesis, cell cycle phase and metabolic functions (Table S1). The PEA protein panel was developed for application across different cellular models, and therefore not all proteins were expected to be detectable at the single cell level in hESCs.

- Sampling time for the velocity analysis in a proof of principle for new technology is wrongly chosen (realized by the authors in the text).
- Velocity vectors going unexpectedly in both directions; it is uncertain if the analysis is fully correct in Figure 2. They explain the arrows from 24h to 0h with 24h being a “transition state”. But how do the cells come to 24h?

We agree that the sampling times are not conducive for a proper velocity analysis. We therefore omitted the analysis from the manuscript. We added the following paragraph to the discussion:

Since SPARC employs the Smart-seq2 protocol, future studies could utilize the sizeable fraction of intronic reads to further dissect gene regulation and expression dynamics. For example, we hypothesize that RNA and protein co-profiling can be used to predict future cell states similar to RNA velocity that employs spliced and un-spliced transcripts^{29, 30}. Updated versions of the Smart-seq2 protocol, such as Smart-seq 2.5 and Smart-seq3, furthermore allow for greater sensitivity and better quantification through the use of UMIs.

- Oct 4 has a half-life time from about 90 min an offset between mRNA and protein level with a sampling time of 24 h cannot be expected.

The protein half-life of POU5F1 (Oct4) is estimated by various studies to be between 4 and 12h in human pluripotent cells (Pan et al. 2016, Lin et al. 2012) and 6.5 to more than 12h in mouse ESCs (Liu et al. 2017, Strebinger et al. 2019). Independent of the RNA half-life, we would therefore expect to see a delay in expression changes. If at a given time point POU5F1 RNA is rapidly repressed and degraded, POU5F1 protein will still be present a considerable time after the reduction in RNA levels. In fact, such substantial delays between RNA and protein levels of pluripotency markers have been shown in differentiation experiments (Rosa and Brivanlou 2011).

References:

- Pan et al. 2016 (“Site-specific Disruption of the Oct4/Sox2 Protein Interaction Reveals Coordinated Mesendodermal Differentiation and the Epithelial-Mesenchymal Transition”) estimate Pou5f1 (Oct4) half-life to be 4-6h (Fig. 6) in NCCIT cells (human, pluripotent)
- Lin et al. 2012 (“Reciprocal Regulation of Akt and Oct4 Promotes the Self-Renewal and Survival of Embryonal Carcinoma Cells”) estimate Pou5f1 (Oct4) half-life to be 6-12h (Fig. 2A) in NCCIT cells (human, pluripotent)
- Liu et al. 2017 “G1 cyclins link proliferation, pluripotency and differentiation of embryonic stem cells”) estimate Pou5f1 (Oct4) half-life to be more than 12h (Suppl. Fig. 4C) in mESCs
- Strebinger et al. 2019 (“Endogenous fluctuations of OCT4 and SOX2 bias pluripotent cell fate decisions”) estimate Pou5f1 (Oct4) half-life to be 7.8 ± 1.3 h (Fig. 1E) in mouse ESCs
- Rosa and Brivanlou 2011 (“A regulatory circuitry comprised of miR-302 and the transcription factors OCT4 and NR2F2 regulates human embryonic stem cell differentiation”) show time courses of hESC differentiation and a substantial delay between RNA and protein (Fig. 1D)

- Why focusing so much on Oct4? For the neuronal development more markers are in foreground. Are those not working?

See earlier comment about the focus on Pou5f1 (Oct4) and Sox2.

- It is not surprising that the network target prediction is working better for Oct4 on the protein level than mRNA since there shouldn't be much mRNA transcripts for a TF so the noise and technical variances between single cells is larger.

The transcription factors profiled here are moderately expressed (see Figure 1) and not in a regime that is dominated by technical noise, especially sampling noise. We estimated 16 RPM to clearly exceed this regime in previous studies using the Smart-seq2 protocol (Tarbier, et al. 2020).

However, we agree that it is difficult to estimate the exact impact of RNA detection noise on these analyses. Interestingly, despite the possible impact of technical noise sources, especially on lowly expressed genes, we can see clear and convincing covariations between individual POU5F1 targets at the RNA level (see figure below), including highly (e.g. HSPD1 and TDGF1) and lowly (e.g. NANOG and ETV4) expressed ones. In contrast, POU5F1 RNA is relatively poorly correlated with these targets, while POU5F1 protein shows high correlation.

If the higher technical variability of RNA measurements would drive our observations of transcription factor target interaction, we should also see lower correlation of targets among each other on the RNA level. Clearly this is not the case, instead there is a clear hierarchy: $\text{target}_{\text{RNA}} \text{ with } \text{TF}_{\text{RNA}} < \text{target}_{\text{RNA}} \text{ with } \text{target}_{\text{RNA}} < \text{target}_{\text{RNA}} \text{ with } \text{TF}_{\text{Protein}}$.

Figure 1: This bar plot shows the number of highly significant ($p < 10^{-9}$) correlations between the gene indicated on the x-axis and the top 10 POU5F1 (OCT4) targets: AK4, BNIP1, EPCAM, ETV4, HLTf, HSPD1, IL17RD, NANOG, RBM47, and TDGF1). Color indicates the RNA expression level. Targets RNA levels themselves show many significant correlations with other targets RNA levels. POU5F1 RNA is only significantly correlated with a single one of the top 10 targets. POU5F1 protein on the other hand correlates significantly with 8 of the top 10 targets. There is no strong relationship between the number of significant correlations and the RNA expression level.

Moreover, the notion that the more reliable detection of protein results from its higher molecule abundance is not contrary to the message of the paper. One argument we aim to make with this

study is the added value of measuring protein versus RNA in single cells, including the lower impact of sampling noise on protein measures. We therefore suggest that the higher correlation of POU5F1 protein with targets has two origins - the closer temporal connection between protein TF and its RNA target, and sampling of abundant protein versus less abundant RNA molecules. In the presented experimental approach, we did not aim to distinguish biological and technical effects, but this should be the aim of future studies.

- Probably very variable amplification biases between the RNA detection (smartseq) and the protein detection (a qPCR)? A synonymous readout would produce much better/ easier to handle datasets.

First, qPCR is a very sensitive detection method. Second, both the mRNA and antibody-associated DNA go through almost equal rounds of PCR amplification before being detected via either sequencing or qPCR, respectively. However, we agree that there would be some benefits to a completely sequencing based readout and hope to develop the method in this direction in the future. We note that even with an unified sequencing readout, the sequencing library preparation for the RNA and protein would still need to be separate due to e.g. sample tagging via PCR versus Tn5 and differences in amplicon size and proportional amount.

Even for CITE-seq, where the sample tags are added in the same way within a droplet, the downstream library preparation is performed in separate tubes, using different PCR primers. This is due to two reasons: First, protein tags are very short in comparison to cDNA from RNA molecules; and, second, the protein molecules are far more abundant than the RNA molecules. Only a small fraction of the protein library is then spiked into the sequencing run.

Finally, it has to be noted, that protein measures will never be completely comparable to RNA measures even when using the same readout. Due to the DNA-conjugated antibody-based detection, the protein values are inherently semi-quantitative and informative for expression changes, variation and covariation, but not for absolute abundance.

- Unclear how the protein level normalization has been done. If the data has been normalized per single cell most of the biological variation has been averaged? Indication for this is that cq values are very similar for all proteins.

Protein normalization is described in the Methods section under “scProtein dataset processing”. In brief, protein data has been normalized within each plate (Cq of each Assay - Cq of an internal control) and adjusted for the negative control (dCq - (lysis buffer mean + 2*SD)). Protein values are not normalized to total protein sums, since total protein amounts can vary depending on, e.g., cell cycle stage.

- Is the reason of low variability of the protein levels due to insensitivity of the qPCR method?

In general quantitative PCR is a very sensitive detection method. A comparison of 100 pooled cells with single cell measures yields a correlation of 0.88 (see figure below). The exact

variability of protein measures is furthermore not determined by the exact percentage of detected molecules as long as it is comparable across individual cells. We do not believe that there are substantial differences in the sensitivity between individual qPCR reactions. Additionally, only a very small fraction of the proteins is close to the detection limit (see Supplementary Figures 2 and 4).

Figure 2: Analysis of protein expression in single cells versus 100 cells
 Agreement of average measured protein concentration levels (**Cq**) in single cells and 100 FACS sorted cells. Each dot represents a measured protein. The measurements of NOTCH1, POUF51, SOX2, CASP3, EPCAM and TP53 are also identified by their names. The Spearman rank correlation coefficient (ρ) between the single cells and the 100 cells is 0.88 (p -value $8.99e-21$). Color of the dots represent the fraction of cells were the protein was considered to be expressed ($Cq > 0$). The lighter the color the higher fraction of cells in which the protein was expressed.

- Cite seq data has a much higher single cell variability, why? Could this indicate that a the normalized qPCR is the problem. If so the entire results to figure 4 is affected.

First, we did not compare protein expression in the same cell line with both SPARC and CITE-seq, therefore we do not know if CITE-seq single cell data is more variable. Second, we show that protein expression across proteins can more or less be variable, not uniformly expressed, and therefore the PEA normalization method is not over-correcting the data.

- Is the low cell number a problem for all analyses?

Previous work with single-cell data, especially with quantitative Smart-seq2 data, has shown that the kind of analyses are feasible with around 100 cells per condition or time point. We therefore believe that the number of cells per se was not a limiting factor.

- It is fine to realize that protein and mRNA levels are not redundant but the technical proof for this is too low. The protein method must be validated with an alternative method like Cite seq for surface proteins.

We agree that it could be informative to have single cell proteomics data from an orthogonal approach. However, we motivate that we present multiple pieces of evidence that strongly support the notion that the protein data is robust and highly quantitative. First, the median expression measures over time scale well with a control of 100 pooled cells. Second, there is a good agreement between estimated pseudotime from RNA and protein measures, the same holds true for general dimensionality reductions. Third, the covariation of RNA levels of transcription factors with transcription factor protein can be seen as a positive control. The abundance of targets has been used in the past to computationally estimate the abundance and activity of transcription factors, and single cell studies have shown that the transcription factor RNA does not correlate with target RNA. Finally, antibodies have been thoroughly tested and validated by us and Olink Proteomics.

We think there is little evidence that the protein measures would be a reflection of sampling noise only. We understand that caution should be applied when assessing new technologies. However, we also believe that we present manifold evidence for the quantitiveness of the protein data as well as the non-redundancy of RNA and protein measures.

In general PEA is a well-established technology that has been benchmarked for its quantitiveness. In this study we show that single-cell measures correlate well ($\rho=0.89$) with measures obtained from pooled 100 cell controls.

Minor:

- Do they use the whole mRNA count for the RNA-protein velocity or only the unspliced mRNA count?

All RNA counts have been used for the RNA-protein velocity. The RNA velocity was performed as usual using unspliced and spliced transcripts respectively. Regardless, we decided to remove this analysis from the manuscript.

- Keyword misleading. There is microfluidics, expect a commercially available platform.

We agree and therefore remove the keyword “microfluidics”. Instead we add “PEA”, which is a necessary technology for the protocol.

- No info about the antibodies and design of the antibody probes – Sequences, washing steps? Unspecific bindings? Unbound antibodies will also hybridize and give signal.

All antibodies have been obtained from Olink and a complete list of proteins IDs can be found in Supplementary Table 1. While we cannot disclose the oligo sequences (as they are property of Olink) all information about individual assays is available on their website: olink.com. In brief, each PEA assay was assessed for sensitivity, cross-reactivity, and appropriate dose-response against cell lysates or recombinant antigens. Assays displaying single- or near-single-cell sensitivity, minimal variation in replicates and no evidence of cross-reactivity were chosen for subsequent analysis.

One strength of PEA and SPARC is that the reaction occurs in solution phase and no wash steps are required after antibodies are added to the cell lysate. Antibody incubations are performed in low volumes followed by a 25x dilution for the DNA extension step. Performing the extension step at high dilution minimizes the extension of antibodies in proximity by chance. As with all antibody methods, there is indeed a background antibody signal, but the signal is stable, calculated for each antibody and every experiment, and is subtracted during the normalization steps.

- These problems are mentioned in Supplemental Results and Figures but kind of glossed over otherwise.
- Fig 1 - A + B Very simplified graphics, almost no info about the antibody PEA method

Figures 1A and 1B are indeed simple overview figures. PEA is an established technology and we refer readers to previously published papers on the matter.

- Fig 4 -Central findings (variability mRNA/protein) are somehow presented very underwhelmingly with very few examples named – why not choose a plot that shows variation for each of their calculated values?

Figure 4A follows the standard visualization of the dependency of mean and variation in single cell experiments. More detailed overviews of the expression profiles (which show variation) can be found in Supplementary Figures 2 and 4. Figure 4B mimics this style of 4A to allow for an easy comparison between RNA and protein variation. We improved the layout of the figure further, but think that its simplistic representation of information is not per se bad. It rather makes the information more accessible to the reader.

For readers that are interested in any particular gene we now include a Supplementary Figure that shows all genes labeled (Suppl. Fig. 8).

- Fig. 1c shows only 4 proteins are shown, Figure S2, and Table S1 summarizes

Reviewer #2 (Remarks to the Author):

The authors report single-cell protein and RNA co-profiling (SPARC) for simultaneously measuring global mRNA and large sets of intracellular protein in single cells. With SPARC analysis they demonstrated that mRNA and protein measurements in single cells provide different and complementary information regarding cell states. However, major concerns for this study lack the novelty for SPARC which simply combined two well-established Smart-seq2 for RNA analysis and PEA for protein analysis as well as sufficient performance evaluation of SPARC.

In recent years there are many methods reported for simultaneous analysis of protein and RNA (Nat Methods 2016, 13, 269; Nat Biotech 2017, 35, 936; Cell Syst 2018, 7, 398; Cell Syst 2018, 6, 531; Nat Commun 2019, 10, 3544) including one study from author group (Cell Reports 2016, 14, 380-389). Systematic performance comparison with at least one existing method is required to validate SPARC in terms of sensitivity, accuracy, reproducibility, and quantitative dynamic range. Modification of current Smart-seq2 method is not novel with incremental improvement in performance.

In principle, SPARC is the combination of two existing protocols. It is nevertheless the very first technology to ever quantify the entire transcriptome as well as dozens of proteins independent of protein localization in the exact same single cells. This combination of technologies has never been successfully executed before and came with its own technical challenges that had to be overcome by the authors. In the table below we compare different recent technologies and show how SPARC outperforms existing competitors.

Reference	Acronym or short description	RNA detection	Protein detection	Major Limitations
Targeted RNA and protein detection, channel limited				
Frei, et al. 2016	PLAYR	Targeted, proximity ligation assay, cytometry (limited by channels, 5 to 10) or CyTOF (limited by channels, 40 to max. 135 genes)	Targeted, antibodies, CyTOF (limited by channels, 40 to max. 135 genes)	Probes needed for RNA detection, need for highly specific antibodies, limited by available channels (both RNA and protein), fixed cells, semi-quantitative ^{1,2,3}
Schulz, et al. 2017	Imaging mass cytometry	Targeted, bDNA (metal-oligo), CyTOF (limited by channels, 40 to max. 135 genes)	Targeted, antibodies (metal-conjugated), CyTOF (limited by channels, 40 to max. 135 genes)	Probes needed for RNA detection, need for highly specific antibodies, limited by available channels (both RNA

				and protein), fixed cells, semi-quantitative ^{1,2,3}
Popovic, et al. 2018	Tagging with fluorescent protein	Targeted, smFISH using bdNA (limited by channels)	Targeted, genetically GFP-tagged proteins (1 gene per experiment, no parallelization), independent quantification using antibodies	Probes needed for RNA detection, Either only 1 protein per experiment (genetics, GFP) or semi-quantitative ^{1,2,3} with need for highly specific antibodies, RNA detection limited by channels, fixed cells
Targeted RNA and protein detection, qPCR-based				
Darmanis, et al. 2016	Combined PEA and qPCR	Targeted, microfluidic qPCR	Targeted, proximity extension assay, microfluidic qPCR	Probes needed for RNA detection, semi-quantitative ¹
Whole transcriptome and targeted cell surface protein detection				
Peterson, et al. 2017	REAP-seq	Droplet microfluidics based scRNA-seq	Targeted, DNA-labeled antibodies, dozens to hundreds of genes, cell surface proteins only	Cell surface proteins only, need for highly specific antibodies, semi-quantitative ¹
Stoeckius, et al. 2017	CITE-seq	Droplet microfluidics based scRNA-seq	Targeted, DNA-labeled antibodies, dozens to hundreds of genes, cell surface proteins only	Cell surface proteins only, need for highly specific antibodies, semi-quantitative ¹
Whole transcriptome and targeted intra-cellular protein detection				
Gerlach, et al. 2019	RAID	Well-plate based scRNA-seq	Targeted, antibodies (RNA-conjugated), few genes	Fixed cells, few proteins only (N=6, up-scaling unclear), semi-quantitative ¹
Reimegård, et al. 2019 (this study)	SPARC	Droplet microfluidics based scRNA-seq	Targeted, proximity extension assay, targeted, many dozens (~100) of genes, easily scalable	Semi-quantitative ¹

Katzenelen-bogen, et al. 2020	INs-seq	Well-plate or droplet microfluidics based scRNA-seq	Targeted, fluorophore-conjugated antibodies, FACS-based quantification (limited by channels), few genes	Fixed cells (improved fixation), few proteins only, limited by available channels (~10), semi-quantitative ¹
1 protein quantification is semi-quantitative due to variable antibody affinity (abundances compare across cells but not across genes) 2 only applies to one of the alternative methods presented 3 RNA quantification is semi-quantitative due to variable probe affinity (abundances compare across cells but not across genes)				

Minor comments:

1. In page 3 it is not clear why 89 proteins out of the panel of 96 proteins had detectable levels in single cells, while only 87 proteins in the 100 cell control. Please explain this.

The difference is based on the detection of cell cycle specific proteins that are expressed in subset of single cells that are in the cognate cell cycle. In the 100 cell control, only a small fraction of cells are in each cycle, the protein signal is diluted and under the limit of detection.

2. In page 11, significant protein loss may occur due to substantial surface absorption when cell lysate containing the protein supernatant is transferred to a new 96-well PCR-plate. Evaluation of protein recovery at the single-cell level for this transfer is needed for precise quantification.

We agree with the reviewer that some level of protein loss can occur due to surface absorption. We have not measured the extent of protein loss in this study, but in a previous publication (Darmanis et al Cell Reports 2016), we showed that split single cell protein measurements are highly reproducible and correlated. Moreover, in this study, we show that 100-cell replicates are highly reproducible. Taken together, we are confident that protein loss is not driving variation in protein expression.

Reviewer #3 (Remarks to the Author):

The authors measured the abundances of mRNAs and proteins in single cells as they undergo differentiation, and used the data to explore the covariation of transcripts and proteins. They also examined the ability to identify the target genes of TFs based on the correlation between TFs and mRNAs and found that protein abundances of TFs are much more predictive of TF target genes. Furthermore, they report that protein expression variation is lower than mRNA variation.

I found the study interesting and the conclusions generally well substantiated. I particularly like the requirement for pairwise protein detection, which ensures higher specificity compared to methods that use a single antibody per protein. I have a few suggestions to further strengthen the conclusions and their description:

1) It seems to me that the major methodological advance of SPARC over Darmanis, S. et al. (Ref. 11) is replacing the PCR readout with Smart-seq2. I think stating this explicitly will be helpful for clarifying the methodological advance.

We agree with this assessment and now specifically highlight this advance in the main text. "mRNA levels were recorded using a modified Smart-seq2 protocol enabling sensitive expression measurements of the full-length transcripts. The use of Smart-seq2 replaces a targeted, multiplexed qPCR approach described previously."

2) Measurement noise can have large and decisive influence on interpreting protein - RNA correlations <https://journals.plos.org/ploscompbiol/article?id=10.1371/journal.pcbi.1005535> The authors should incorporate reliabilities in their estimates or at least acknowledge the influence of noise. Note that measurement noise might contribute to the higher correlations for the pseudotime trends since some of the noise is averaged and reduced.

We agree that differences in technical variation can impact these analyses and we now highlight this limitation in the Discussion. It is, however, interesting to notice that we observe clear covariations between individual targets on the RNA level, including highly and lowly expressed ones, but specifically not with the factor itself (see comments to reviewer 1). This lends evidence to the notion that biological effects such as covariation as a result of transcription factor co-targeting can be seen on the RNA level, but that target RNAs for biological reasons covary with the protein of transcription factor only.

Text added:

Results from single-cell mass spectrometry (Specht, et al. 2019) support these results. We note that our measurements are not corrected for technical measurement noise, and that incorporating reliability estimates for RNA and protein measurements may provide an even more acute view of mRNA-protein relationships (Franks et al 2019).

3) The lower variability of the protein levels seems consistent with our observations of lower protein variability in Figure 5 and Figure S3 of <https://www.biorxiv.org/content/10.1101/665307v3>. These observations use different data and analysis and mutually support each other. The author may want to discuss this connection.

We thank the reviewer for pointing out the interesting and relevant study. We have referred to the publication in the discussion.

4) This description in the introduction is a bit simplistic

"While bulk sample measurements generally report good correlation between mRNA and protein expression, little is known about the extent of correlation at cellular resolution in systems at steady-state or undergoing a dynamic transition."

Some of the referenced studies reported much larger correlations than other referenced studies, and some of these differences are due to considering or not considering measurement noise as well as to computing the correlations across genes or across conditions. The authors may want to be a bit more specific in their description and what is meant by a "good correlation".

We agree with the reviewer that our statement regarding correlation was over simplistic. We accordingly removed the sentence as we do not think that a summary of a fairly extensive and at times contradictory body of work is pertinent in the introduction.

Sentenced removed:

~~While bulk sample measurements generally report good correlation between mRNA and protein expression^{2,4,7}, little is known about the extent of correlation at *cellular resolution* in systems at steady-state or undergoing a dynamic transition.~~

5) The paper is clearly written and generally easy to follow. At some places, it can benefit from replacing qualitative statements with more exact quantitative statements, e.g., "large sets of intracellular protein" with "sets of about 100 intracellular proteins".

While the set we profile contains close to 100 proteins, the method itself can be scaled to measure more. However, we clarified the number of proteins measured at different points throughout the text.

Generally, I find this paper strong. After incorporating the suggestions listed above, it likely will become one of the strongest papers in the field. SPARC makes very significant improvements over competing methods such as CITE-seq and REAP-seq, and its application demonstrates important results, such as the higher predictive potential of protein measurements for TF-targets.

Nikolai Slavov

Reviewers' Comments:

Reviewer #1:

Remarks to the Author:

The manuscript can be accepted as is.

Reviewer #3:

Remarks to the Author:

The authors have addressed my comments.

Please find attached detailed responses to the reviewers.

REVIEWERS' COMMENTS:

Reviewer #1 (Remarks to the Author):

The manuscript can be accepted as is.

No additional rebuttal required.

Reviewer #3 (Remarks to the Author):

The authors have addressed my comments.

No additional rebuttal required.